# U2UData: A Large-scale Cooperative Perception Dataset for Swarm UAVs Autonomous Flight

### Tongtong Feng
Department of Computer Science and
Technology, Tsinghua University
Beijing, China
fengtongtong@tsinghua.edu.cn

### Xin Wang*
Department of Computer Science and
Technology, BNRist, Tsinghua
University, Beijing, China
xin_wang@tsinghua.edu.cn

### Feilin Han
Department of Film and Television
Technology, Beijing Film Academy
Beijing, China
hanfeilin@bfa.edu.cn

### Leping Zhang
Department of Film and Television
Technology, Beijing Film Academy
Beijing, China
zhangleping@mail.bfa.edu.cn

### Wenwu Zhu*
Department of Computer Science and
Technology, BNRist, Tsinghua
University, Beijing, China
wwzhu@tsinghua.edu.cn

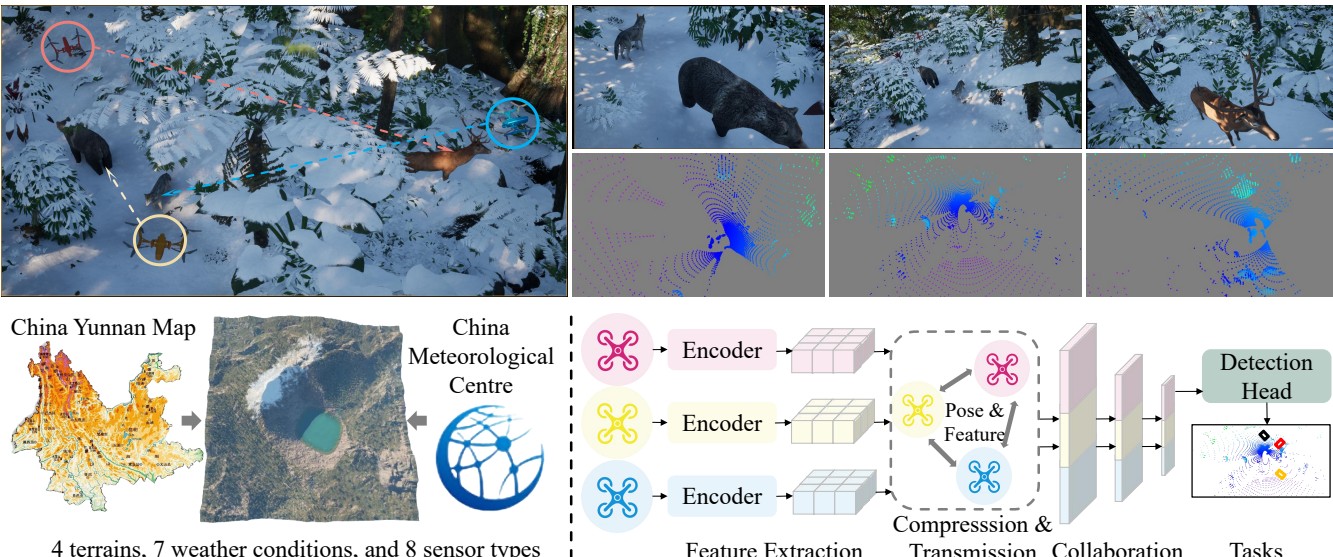

**Figure 1: U2UData is collected by performing swarm UAVs autonomous flight tasks in the U2USim environment. Top left: Swarm UAVs autonomous flight task, where each UAV protects an animal based on the arrow. Top right: First-person views and LiDAR images of each UAV. Bottom left: U2USim, a real-world mapping swarm UAVs simulation environment. Bottom right: Swarm UAVs cooperative perception benchmark.**

## Abstract

Modern perception systems for autonomous flight are sensitive to occlusion and have limited long-range capability, which is a key bottleneck in improving low-altitude economic task performance.

*Corresponding Authors. BNRist is the abbreviation of Beijing National Research Center for Information Science and Technology.

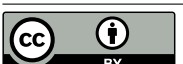

*MM '24, October 28-November 1, 2024, Melbourne, VIC, Australia*
© 2024 Copyright held by the owner/author(s).
ACM ISBN 979-8-4007-0686-8/24/10
https://doi.org/10.1145/3664647.3681151

Recent research has shown that the UAV-to-UAV (U2U) cooperative perception system has great potential to revolutionize the autonomous flight industry. However, the lack of a large-scale dataset is hindering progress in this area. This paper presents U2UData, the first large-scale cooperative perception dataset for swarm UAVs autonomous flight. The dataset was collected by three UAVs flying autonomously in the U2USim, covering a 9 km$^2$ flight area. It comprises 315K LiDAR frames, 945K RGB and depth frames, and 2.41M annotated 3D bounding boxes for 3 classes. It also includes brightness, temperature, humidity, smoke, and airflow values covering all flight routes. U2USim is the first real-world mapping swarm UAVs simulation environment. It takes Yunnan Province as the prototype and includes 4 terrains, 7 weather conditions, and 8 sensor types. U2UData introduces two perception tasks: cooperative 3D

object detection and cooperative 3D object tracking. This paper provides comprehensive benchmarks of recent cooperative perception algorithms on these tasks.

## CCS Concepts

• **Computing methodologies** → **Cognitive robotics**; *Cooperation and coordination*; Robotic planning.

## Keywords

Cooperative Perception; Swarm UAVs; Autonomous Flight; Dataset; Simulation Environment

**ACM Reference Format:**
Tongtong Feng, Xin Wang, Feilin Han, Leping Zhang, and Wenwu Zhu. 2024. U2UData: A Large-scale Cooperative Perception Dataset for Swarm UAVs Autonomous Flight. In *Proceedings of the 32nd ACM International Conference on Multimedia (MM '24), October 28-November 1, 2024, Melbourne, VIC, Australia.* ACM, New York, NY, USA, 9 pages. https://doi.org/10.1145/3664647.3681151

## 1 Introduction

Perception is critical in the autonomous flight task of Unmanned Aerial Vehicles (UAVs) for accurate navigation and safe planning[3, 4]. The recent development of deep learning brings significant breakthroughs in various perception tasks such as 3D object detection[5], object tracking[6], and semantic segmentation[7]. However, single-UAV vision systems still suffer from many real-world challenges, such as occlusion and short-range perception capability[1, 2, 8], which can cause catastrophic accidents and are key bottlenecks in improving low-altitude economic task performance. The shortcomings are mainly due to the limited field-of-view of individual UAVs, resulting in an incomplete understanding of the environment.

A growing interest and recent advances in cooperative perception systems[9–12] have enabled a new paradigm that can potentially overcome the limitations of single-UAV perception. By leveraging UAV-to-UAV (U2U) technologies, multiple connected and automated UAVs (UAVs) can communicate and share captured sensor information simultaneously. As shown in Figure 1, swarm UAVs flying autonomously in a dynamic open environment, the ego UAV (blue) struggles to perceive the tracking object (deer) due to leaf obstruction and snow interference. Incorporating the multimodal features of the nearby UAVs (yellow or red) can further distribute multiple tasks and achieve greater flexibility, robustness, and perceptual range, leading to significant advantages in harsh and complex environments.

However, despite the great promise, cooperative perception is mainly focused on the vehicles and ignores the UAV literature, which remains challenging to validate U2U perception in real-world scenarios due to the lack of public datasets. Existing U2U cooperative perception datasets, as shown in Table 1, CoPerception-UAVs[1] and CoPerception-UAVs+[2] rely on open-source simulators such as AirSim[13] and CARLA[14] and consider only 1 terrain, 1 weather, and 1 to 2 sensor types; they collect datasets using fixed altitude and consistent or fixed formation mode. In real-world scenarios, compared to autonomous driving, autonomous flight has more freedom, faces more complex environments, and is more susceptible to the influence of temperature, humidity, and airflow due to its

smaller size. Obviously, there will be a clear domain gap between existing synthetic data and real-world data. As a result, models trained on these datasets may not generalize well to realistic flight situations.

To further advance innovative research on U2U cooperative perception, 1) we present a large-scale cooperative perception dataset (U2UData) for swarm UAVs autonomous flight. It is collected by three UAVs flying autonomously in the U2USim, covering a 9 $km^2$ flight area, comprising 315K LiDAR frames, 945K RGB and depth frames, 2.41M annotated 3D bounding boxes for 3 classes, and including brightness, temperature, humidity, smoke, airflow values covering all flight paths. Compared with fixed altitude and consistent or fixed formation mode flying, autonomous flight can more comprehensively explore harsh and complex environments, allowing perception models to achieve higher flexibility and stronger robustness. 2) Brightness, temperature, humidity, smoke, and airflow modalities can greatly affect UAV flight control. However, they are closely related to terrain and meteorology, and have strong interactions with each other, making them difficult to collect. We build the first real-world mapping swarm UAVs simulation environment, U2USim, including 4 terrains, 7 weather conditions, and 8 sensor types. It takes Yunnan Province as the prototype and uses the real meteorological data of Yunnan Province collected by the China Meteorological Center[1] to map the simulation environment based on longitude and latitude. It enables UAVs to perceive the world like humans and make optimal decisions. 3) U2UData introduces two perception tasks: cooperative 3D object detection and cooperative 3D object tracking. 4) We have provided 8 state-of-the-art cooperative perception algorithms for benchmarking, and will make all the data, benchmarks, and models publicly available worldwide.

Our contributions can be summarized as follows:

- **Dataset.** We present U2UData, the first large-scale cooperative perception dataset for swarm UAVs autonomous flight.
- **Simulation.** We build U2USim, the first real-world mapping swarm UAVs simulation environment, taking Yunnan Province as the prototype, including 4 terrains, 7 weather conditions, and 8 sensor types.
- **Benchmark.** We introduce two cooperative perception tasks, including cooperative 3D object detection and cooperative 3D object tracking, and provide comprehensive benchmarks with 8 SOTA models. The results show the effectiveness of U2UData in multiple tasks.

## 2 Related Work

**Cooperative perception.** Due to the inherent limitations of an agent's own sensors (e.g., camera/LiDAR), occlusions, sensor degradation or failure, and long-range perception are extremely challenging for single-agent systems, with potentially disastrous consequences in harsh and complex environments. Cooperative perception systems can learn how to share information among multiple agents to perceive the environment better than individually. Existing cooperative perception approaches can be roughly divided into three categories: (1) Early Fusion[21]. Agents transmit raw sensor data directly to other collaborators, and the ego agent makes task decisions based on the fused raw data, which preserves complete

---

[1]https://data.cma.cn/

**Table 1: A detailed comparison between autonomous flight-related datasets. - indicates that specific information is not provided. DF: Discipline formation mode, where swarm UAVs keep a consistent and relatively static array; FF: Fixed formation mode, where each UAV navigates independently with a fixed path; AF: Autonomous formation mode, where each UAV flies autonomously. D: 3D object detection; T: 3D object tracking; S: semantic segmentation.**

| Datasets | Year | Real data | Simulation | UAVs | Altitudes | Formation | Sample | Terrains | weather | Sensors | Tasks |
|---|---|---|---|---|---|---|---|---|---|---|---|
| CoPerception-UAVs[1] | 2022 | - | AirSim + Carla | 5 | Fixed | DF, FF | 4s | 1 | 1 | 1 | D, S |
| CoPerception-UAVs+[2] | 2023 | - | AirSim + Carla | 10 | Fixed | DF, FF | 4s | 1 | 1 | 2 | D, S |
| **U2UData (our)** | 2024 | China | U2USim | 3 | [56.6, 3000] | DF, FF, AF | 0.03s | 4 | 7 | 8 | D, S, T |

**Table 2: A detailed comparison between simulators related to autonomous flight. H: High; M: Medium; L: Low. U2USim takes Yunnan Province as the prototype and uses the real meteorological data of Yunnan Province collected by the China Meteorological Center to map the simulation environment based on longitude and latitude. U2USim includes dynamic temperature, humidity, smoke, and airflow sensor information.**

| Simulator | FightGear[15] | XPlan[16] | Jmavsim[17] | Gazebo[18] | AirSim[13] | RflySim[19] | Isaac Sim[20] | **U2USim (Our)** |
|---|---|---|---|---|---|---|---|---|
| Open Source | - | ✓ | ✓ | ✓ | ✓ | ✓ | ✓ | ✓ |
| ROS | - | - | ✓ | ✓ | ✓ | ✓ | ✓ | ✓ |
| Sensor Output | - | - | - | ✓ | ✓ | ✓ | ✓ | ✓ |
| Physical Collision | - | - | - | ✓ | ✓ | ✓ | ✓ | ✓ |
| Records | - | - | - | - | ✓ | ✓ | ✓ | ✓ |
| Weather | - | - | - | - | ✓ | ✓ | ✓ | ✓ |
| Real Data | - | - | - | - | - | - | Digital Twin | Mapping |
| Altitude | - | - | - | - | - | - | ✓ | ✓ |
| Temperature Sensor | - | - | - | - | - | - | - | ✓ |
| Humidity Sensor | - | - | - | - | - | - | - | ✓ |
| Smoke Sensor | - | - | - | - | - | - | - | ✓ |
| Airflow Sensor | - | - | - | - | - | - | - | ✓ |
| Fidelity | L | H | L | M | H | H | H | H |
| Richness | L | H | L | M | H | H | H | H |

*

information but requires a large bandwidth. (2) Late Fusion[22]. Each agent makes task decisions using its sensor data and delivers the decision results to others. The ego agent applies Non-maximum suppression to produce the final outputs, which maintain small transmission bandwidths but cannot achieve deep fusion of sensor information. (3) Intermediate fusion[1, 7, 9–12]. Adjacent UAVs utilize a neural feature extractor to derive intermediate features, which are then compressed and sent to the ego UAV for cooperative feature fusion. This method maximizes the benefits of both early and late fusion and is well-suited for extensive deployment scenarios. However, despite its great promise, cooperative perception mainly focuses on vehicles and ignores the UAV literature, which remains challenging to validate U2U perception in real-world scenarios due to the lack of public datasets.

**Cooperative perception datasets.** Collecting datasets[23–29] is crucial for advancing algorithmic research. Public cooperative perception datasets have significantly accelerated progress in autonomous driving technologies in recent years. For instance, OPV2V[30] introduced the first 3D cooperative perception dataset, leveraging CARLA[14] and OpenCDA[31] joint simulation environment. DAIR-V2X pioneered real-world datasets for cooperative detection. V2x-seq[32] developed the initial sequential dataset for vehicle-infrastructure cooperative perception and forecasting. V2X-Sim[33] further explored vehicle-to-everything perception feasibility using

synthesized data from the CARLA simulator[34]. V2V4Real[35] established the first large-scale real-world dataset for vehicle-to-everything cooperative perception. However, cooperative perception datasets in autonomous flight scenarios remain scarce. Existing U2U cooperative perception datasets, such as CoPerception-UAVs[1] and CoPerception-UAVs+[2], rely on open-source simulators like AirSim[13] and CARLA[14], featuring limited terrain, weather, and sensor types. These datasets collect data at fixed altitudes and in consistent or fixed formation modes. In contrast to autonomous driving, UAVs' autonomous flight presents greater freedom, encounters more complex environments, and is more influenced by natural weather due to their smaller size. Hence, there exists a notable domain gap between existing synthetic data and real-world data, potentially limiting the generalization of models trained on these datasets to realistic flying scenarios. In this paper, we introduce the first large-scale cooperative perception dataset for swarm UAVs autonomous flight.

## 3   U2USim Environment

A simulator for swarm-UAVs must be able to more realistically simulate dynamic physical characteristics[20] (such as collision); sensors such as IMU[15], camera[16, 17], GPS[18], lidar[13] etc; and interaction with the ROS ecosystem[19]. In real-world scenarios,

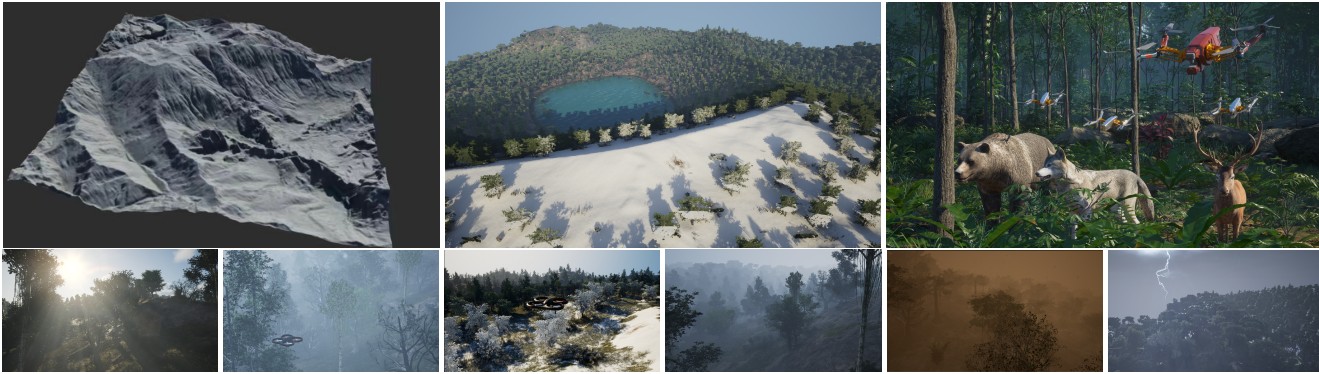

**Figure 2: U2USim overview. The upper left is U2USim's topographic map, the upper middle is U2USim's vegetation conditions, and the upper right is U2USim's tasks. Below are 6 weather views, from left to right: sunny, rain, snow, fog, sandstorm, and thunder. Since wind is invisible, there is no collection view.**

UAVs are extremely sensitive to the effects of temperature, humidity, and airflow due to their small size. Existing UAV simulators, as shown in Table 2, can visually realize digital twins[20] of the real world through GPU rendering, but it is difficult to provide modal information other than visual and LiDAR modalities. Many existing studies manually set values based on experience for temperature, humidity, brightness, smoke, and airflow modalities. However, because sensors are closely related to terrain and meteorology, and have strong interactions between modalities, the model trained by existing simulators is difficult to deploy in the real world.

In this section, we build U2USim, the first real-world mapping swarm-UAVs simulation environment, taking Yunnan Province as the prototype, including 4 terrains, 7 weather conditions, and 8 sensor types. We use actual meteorological data from Yunnan Province collected by the China Meteorological Center to map the simulation environment based on longitude and latitude. U2USim can not only significantly reduce the domain gap, but UAVs can also use it to perceive the world and make optimal decisions like humans.

### 3.1 U2USim Design

**U2USim map.** Yunnan Province is located in southwest China, with high terrain in the northwest and low terrain in the southeast; the elevation range is [76.4, 6740]m. Yunnan Province has a subtropical plateau monsoon climate with significant vertical climate characteristics. The temperature changes vertically with the terrain, so the map of Yunnan Province is an excellent real-world scene for constructing a simulation environment.

As shown in Figure 2, U2USim uses Unreal Engine (UE) 5.2[2] to construct a scaled-down 3km*3km simulated environment map based on the map of Yunnan Province. U2USim includes 4 types of terrain: mountains, hills, plains and basins. The elevation range is [56.6, 3000]m. Based on the vegetation distribution in Yunnan, 58 types of original forest vegetation assets were constructed, and more than 15 superposition methods were used to combine them, including epiphytic growth, diagonal staggered growth, and so on. Among them, the leaves of each plant will dynamically change with wind, rain, snow, and other weather conditions. U2USim takes

Yunnan Province as a prototype, including 4 terrains, 7 weather conditions, and 8 sensor types. It uses the real meteorological data of Yunnan Province collected by the China Meteorological Center to map the simulation environment based on longitude and latitude.

**7 weather conditions.** The weather system is designed to simulate the dynamic weather environment in Yunnan Province. It consists of two parts: the practical code and the artistic performance. To create these impressive weather effects, 19 materials and 78 textures were used. To imitate reality, the simulation includes effects that occur from the sky to the ground, such as wet leaves and muddy floors after heavy rain in Yunnan Province. In addition, there are eight highly customized particle systems to simulate rain, snow, dust, and fog at specific positions within the simulation environment. All of these effects are primarily based on Unreal Blueprints[3], which can reduce the time required to modify complex code. This allows visual designers to participate more in the programming process and easily change the value for realistic modification.

**8 sensor types.** The sensors in U2USim are based on the Unreal Engine, which allows cross-platform compatibility and provides accurate data due to the engine's strong physics calculation capabilities. The LiDAR sensor primarily uses ray detection, and with the powerful engine, it can save a lot of time in tracking and calculating each ray. All image-based data, including depth and flare, are generated using Unreal rendering techniques. The flare image is created through the ID segmentation process, where objects are rendered using mapped temperature colors and output. The depth image is captured during the depth test process. The Luminance sensor uses a camera on the drone to capture live textures and calculate the average brightness and smoke concentration of each pixel. The brightness function is based on the color stimulus curve[36] of the human eye. The smoke concentration function is based on the Laplacian gradient sum algorithm. To enhance the credibility of the data captured by the sensors, including temperature, humidity and airflow, weather data from Yunnan Province was used to map into the simulator environment (as shown in subsection 3.2).

These sensors are installed on the multirotor to explore the simulator map and collect data at 0.02-second intervals, which

---

[2]https://www.unrealengine.com/en-US/unreal-engine-5

[3]https://www.unrealengine.com/marketplace/en-US/content-cat/assets/blueprints

can be customized using a JSON settings file. Sensor mounting is also customizable; users can edit the JSON file to customize their own multirotor by selecting practical sensors from 9 templates and designing the sensor accuracy and return rate.

## 3.2 Real-world mappinng

We use the real meteorological data of Yunnan Province collected by the China Meteorological Center to map the simulation environment based on longitude and latitude. First, the temperature, humidity, pressure, wind speed, and wind direction data collected by weather stations in Yunnan Province from March 5 to 10, 2024, released by the National Meteorological Information Center are downloaded. The real sensor information is then mapped into the U2USim based on the latitude and longitude information of the real map. Among them, temperature and humidity are scalars, and missing values are filled by the moving average method (interval 5m). For wind speed and direction, assuming that the latitudinal wind speed of a location on March 5, 2024 is $u_h$ $m/s$ and the longitudinal wind speed is $v_h$ $m/s$, the formulas for the synthesized wind speed $v_h$ and wind direction $\theta_h$ are as follows:

$$V_h = \sqrt{u_h^2 + v_h^2} \tag{1}$$

$$\alpha = \arctan\left(\frac{u_h}{v_h}\right) \times \frac{180}{\pi} \tag{2}$$

$$\theta_h = \begin{cases} 0 & (u_h = 0, v_h \leq 0) \\ 90 & (u_h < 0, v_h = 0) \\ 270 & (u_h > 0, v_h = 0) \\ 180 + \alpha & (v_h > 0) \\ 360 + \alpha & (u_h > 0, v_h < 0) \\ \alpha & (u_h < 0, v_h < 0) \end{cases} \tag{3}$$

where $\alpha$ is the angular parameter. The equations for latitudinal wind speed $u_h$ and longitudinal wind speed $v_h$ for missing values are as follows:

$$u_h = \frac{u_1 \times (H_2 - H_1) + \sum_{i=2} u_i \times (H_{i+1} - H_{i-1}) + u_n \times (H_n - H_{n-1})}{2 \times (H_n - H_1)} \tag{4}$$

$$v_h = \frac{v_1 \times (H_2 - H_1) + \sum_{i=2} v_i \times (H_{i+1} - H_{i-1}) + v_n \times (H_n - H_{n-1})}{2 \times (H_n - H_1)} \tag{5}$$

where $H_h$ is the potential height of isobars (obtained by sliding average of neighboring sampling points based on distance), $n$ is the number of isobars, and $i$ is the isobar at the missing value location, and the synthetic wind speed and direction are calculated according to Eqs. (1-3).

## 4 U2Udata Dataset

To expedite progress in U2U cooperative perception, we propose U2UData, the large-scale, autonomously flying, real2sim, multi-modal dataset with different weather scenarios. This dataset is meticulously annotated with 3D bounding boxes to facilitate research on swarm-UAVs cooperative perception.

**Sensor setup.** We collect the U2UData using three UAVs flying autonomously in the U2USim. All UAVs are equipped with 5 RGBD cameras (front, back, left, right, and bottom), a 64-LiDAR sensor (top), 1 brightness, temperature, humidity, and smoke sensor (bottom), 2 airflow sensors (back and right), and GPS/IMU systems.

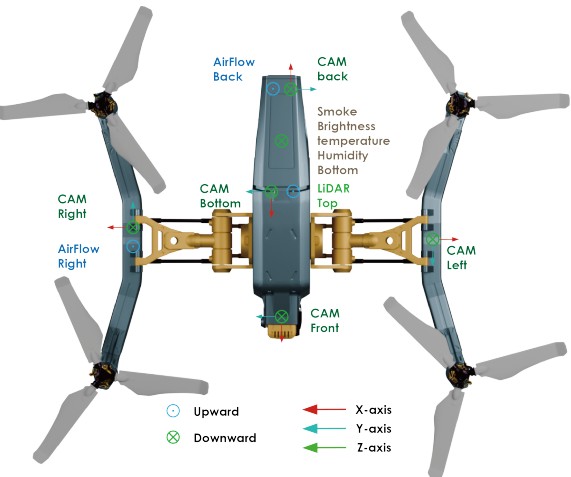

**Figure 3: Sensor setup for our data collection platform.**

**Table 3: Sensor specifications for each UAV.**

| Sensors | Details |
|---|---|
| 5x Camera | RGBD, 1920x1080, FOV: 90, 30Hz |
| 1x LiDAR | 64 channels, 1 M points per second, 200m capturing range, 30° to 30° vertical FOV, 180° to 180° horizontal FOV, ±3cm error, 10 Hz |
| 2x Airflow sensor | latitudinal wind speed, longitudinal wind speed |
| Other Sensors | 1x brightness, 1x temperature, 1x humidity, and 1x smoke sensor |
| GPS & IMU | Odometry |

Figure 3 illustrates the sensor layout configuration, while Table 3 provides a detailed breakdown of the parameters.

**Formation mode.** The existing UAV flight modes of data collection mainly adopt discipline formation and fixed formation modes. As shown in Figure 4, in discipline formation mode, the swarm UAVs maintain a consistent and relatively static array, which has the same height, fixed spacing, and the same speed. In fixed formation mode, each UAV navigates independently with a fixed path, having the same height and speed. However, in a dynamic and open real-world environment, such as UAV delivery tasks, it is difficult for swarm UAVs to maintain a fixed height, speed, and path. In U2UData, we have adopted a new autonomous formation mode in which each UAV flies autonomously. Compared with fixed altitude and discipline or fixed formation mode flying, autonomous flight can more comprehensively explore harsh and complex environments, allowing perception models to achieve higher flexibility and stronger robustness.

**Data collection.** For the 7 representative weather conditions in U2USim, we manually selected 100 scenarios for each weather condition, including different altitudes and different vegetation coverage, covering a 9 km² flight area. For each scenario, we collected 15 seconds of swarm UAVs cooperative perception data. As shown in Table 4, we sample the image frames at 30Hz, comprising a total 945K RGB frames and 945K depth frames. We collect 315K liDAR frames at a sampling frequency of 10Hz, including 2.41M annotated

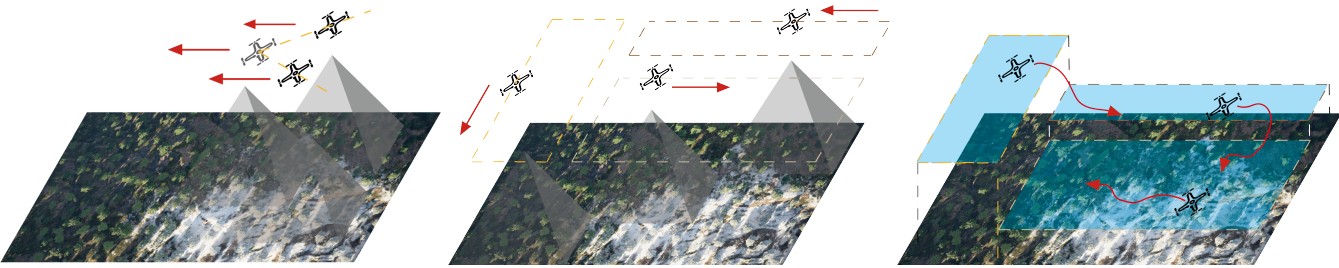

**Figure 4: Three types of swarm-UAVs formation. Left: Discipline formation mode, where swarm-UAVs keep a consistent and relatively static array; Medium: Fixed formation mode, where each UAV navigates independently with a fixed path; Right: Autonomous formation mode, where each UAV flies autonomously.**

**Table 4: A detailed data comparison between autonomous flight-related datasets.**

| Datasets | Year | RGB sensor | | Depth | LiDAR | | Airflow | Brightness | Temperature | Humidity | Smoke |
| | | RGB | Resolution | | LiDAR | 3D boxes | | | | | |
|---|---|---|---|---|---|---|---|---|---|---|---|
| CoPerception-UAVs[1] | 2022 | 131.9K | 800*450 | - | - | 1.94M | - | - | - | - | - |
| CoPerception-UAVs+[2] | 2023 | 52.76K | 800*450 | 52.76K | - | - | - | - | - | - | - |
| U2UData | 2024 | 945K | 1920*1080 | 945K | 315K | 2.41M | 1.89M | 945K | 945K | 945K | 945K |

3D bounding boxes for 3 classes. The 945K brightness, 945K temperature, 945K humidity, 945K smoke, and 1.89M airflow values were collected through real-world mapping according to the acquisition coordinates of each image.

**3D bounding boxes annotation.** For annotating 3D bounding boxes on the gathered LiDAR data, we utilize SusTechPoint[37], a robust open-source labeling tool. We also manually refine the annotations. There are a total of three object classes, including bear, deer, and wolf. For each object, we annotate its 3D bounding box with 7 degrees of freedom, encompassing its location (x, y, z) and rotation (expressed as quaternions: w, x, y, z). The location (x, y, z) corresponds to the center of the bounding box. These 3D bounding boxes are annotated separately based on the global coordinate system of each UAV. This approach enables the sensor data from each UAV to be treated independently as a single-agent detection task. We initialize the relative pose of the two UAVs for each frame using positional information provided by the GPS on both UAVs.

**Data usage.** In total, U2UData has 315K LiDAR frames, 945K RGB and depth frames, 2.41M 3D bounding boxes, 1.89M airflow value, 945K brightness, temperature, humidity, and smoke values. We randomly divide it into training sets/validation sets/test sets according to the ratio of 0.7/0.15/0.15. It can greatly facilitate the credibility of algorithm performance compared to different papers.

## 5 Tasks

### 5.1 Cooperative 3D Object Detection

Several prior works have demonstrated the efficacy of cooperative perception utilizing LiDAR sensors[1, 2, 8]. Nevertheless, whether, when, and how this U2U cooperation using other modalities perception systems has few explored works. In this paper, each UAV first extracts LiDAR modal features through feature encoders and transmits them to adjacent UAVs. In the future, multimodal collaborative perception algorithms can be designed based on this dataset.

**Challenge.** In contrast to the single-UAV detection task, cooperative detection presents several domain-specific challenges:

- *Bandwidth limitation.* Conventional U2U communication technologies often operate within narrow bandwidth constraints[7, 9, 10] and cooperative detection algorithms need to balance between precision and the bandwidth cost carefully.
- *Location error.* Due to environmental disturbances, inherent errors[38] in the adjacent UAVs' relative poses will lead to coordinate system mapping errors.
- *Asynchronicity.* Heterogeneous UAVs[2] are inevitable and the distance between UAVs is variable. Different sensor sampling frequencies and transmission latencies will cause UAVs asynchronous.

**SOTA methods.** We evaluate four categories of outstanding multi-agent embodied cooperative perception methods:

- *No Fusion.* We exclusively utilize the multimodal data from the ego UAV for object detection, establishing this approach as the baseline strategy.
- *Early Fusion.* Each UAV will transmit raw multimodal information directly to other collaborators[21], and the ego UAV aggregates all multimodal features to its own feature map.
- *Late Fusion.* Each UAV makes task decisions utilizing its multimodal information and delivers the decision results to others[22], and the ego UAV applies non-maximum suppression to produce the final results.
- *Intermediate Fusion.* Adjacent UAVs employ a neural feature extractor to derive intermediate features, which are subsequently compressed and broadcasted to the ego UAV for cooperative feature fusion. It can fully exploit the advantages of early and late fusion and is suitable for large-scale deployment. We benchmark some outstanding intermediate fusion methods, including When2Com[12], DiscoNet[11], V2VNet[9], V2X-ViT[10], CoBEVT[7], and Where2com[1].

**Table 5: Synchronous cooperative 3D object detection benchmark.**

| Method | Comm | AP@IoU=0.5 | | | | AP@IoU=0.7 | | | |
|---|---|---|---|---|---|---|---|---|---|
| | | Overall | 0-30m | 30- 50m | 50-100m | Overall | 0-30m | 30-50m | 50-100m |
| No Fusion | 0 | 23.04 | 59.69 | 20.65 | 1.82 | 14.00 | 37.73 | 10.43 | 0.30 |
| Late Fusion | 0.003 | 45.73 | 70.59 | 40.89 | 22.65 | 21.40 | 33.44 | 11.09 | 2.84 |
| Early Fusion | 0.82 | 49.18 | 74.32 | 38.41 | 26.80 | 35.14 | 36.74 | 15.22 | 7.73 |
| When2Com[12] | 0.12 | 47.83 | 65.69 | 37.24 | 29.54 | 37.37 | 34.84 | 16.71 | 9.70 |
| DiscoNet[11] | 0.13 | 53.65 | 70.75 | 44.06 | 33.74 | 52.36 | 43.18 | 19.53 | 12.10 |
| V2VNet[9] | 0.13 | 59.00 | 71.32 | **47.68** | 35.75 | 48.75 | 36.22 | 22.90 | **13.64** |
| V2X-ViT[10] | 0.12 | 56.15 | 75.58 | 43.99 | 35.73 | 50.17 | **46.95** | 19.19 | 9.22 |
| CoBEVT[7] | 0.13 | 60.37 | 76.85 | 44.81 | 41.59 | 52.58 | 43.08 | 22.44 | 9.95 |
| Where2com[1] | 0.13 | **60.77** | **77.87** | 46.71 | **42.03** | **54.71** | 45.74 | **23.75** | 11.85 |

**Table 6: Asynchronous cooperative 3D object detection benchmark.**

| Method | Comm | AP@IoU=0.5 | | | | AP@IoU=0.7 | | | |
|---|---|---|---|---|---|---|---|---|---|
| | | Overall | 0-30m | 30- 50m | 50-100m | Overall | 0-30m | 30-50m | 50-100m |
| No Fusion | 0 | 23.04 | 59.69 | 20.65 | 1.82 | 14.00 | 37.73 | 10.43 | 0.30 |
| Late Fusion | 0.003 | 45.19 | 64.26 | 36.73 | 18.29 | 15.66 | 26.22 | 13.70 | 1.44 |
| Early Fusion | 0.82 | 47.48 | 70.72 | 29.59 | 23.69 | 18.15 | 38.28 | 8.68 | 3.82 |
| When2Com[12] | 0.12 | 46.61 | 66.59 | 30.68 | 17.85 | 16.92 | 30.35 | 10.86 | 0.53 |
| DiscoNet[11] | 0.13 | 52.50 | 72.84 | 40.32 | 19.22 | 21.88 | 39.79 | 16.43 | 0.77 |
| V2VNet[9] | 0.13 | 50.05 | 71.80 | 40.34 | 21.96 | 21.22 | 34.54 | **20.11** | 4.53 |
| V2X-ViT[10] | 0.12 | 49.47 | 75.85 | 35.37 | 18.03 | **24.51** | **44.67** | 14.82 | 3.36 |
| CoBEVT[7] | 0.13 | 50.62 | **76.31** | 36.73 | 22.69 | 22.30 | 40.56 | 17.21 | **5.37** |
| Where2com[1] | 0.13 | **56.45** | 74.84 | **40.45** | **23.94** | 20.28 | 41.83 | 17.42 | 4.26 |

**Evaluation.** The evaluation area extends by [-100, 100]m in both the x and y directions relative to the ego UAV. Our evaluation metric for UAV detection performance is the Average Precision (AP) at Intersection over Union (IoU) thresholds of 0.5 and 0.7. To assess transmission efficiency, we utilize the average bandwidth, which quantifies the transmitted data size as specified by the algorithm. Consistent with [2, 39], we assess all models under two conditions: 1) Synchronous setting: All UAV feature fusion modules do not consider transmission delay and UAV heterogeneity. 2) Asynchronous setting: All UAV data transmission delays are randomly selected between [0, 150]ms and are interfered by the sampling frequencies of heterogeneous UAV sensors.

## 5.2 Cooperative 3D Object Tracking

**Challenge.** Object tracking is a process in which the algorithm tracks the movement of an object. The goal is to track the motion of one or more objects, over multiple frames. This motion estimate changes over time, typically represented by their location, velocity, and acceleration at specific times. In contrast, object detection locates objects within individual frames without establishing associations over time. In cooperative 3D object tracking, multimodal information must be shared among multiple agents while adhering to communication bandwidth constraints to jointly track 3D targets. This task, which differs from traditional single-agent 3D target tracking, presents challenges such as multi-view information fusion between agents, multimodal data fusion, spatio-temporal asynchrony, and limited communication capabilities.

**SOAT methods.** In this subsection, we used 9 SOAT cooperative perception algorithms (as shown in Subsection 5.1) to verify the performance of the 3D object tracking task in the U2UData dataset, including No Fusion, When2Com[12], DiscoNet[11], V2VNet[9], V2X-ViT[10], CoBEVT[7], and Where2com[1].

**Evaluation.** We utilize the same evaluation metrics as outlined in [40] for object tracking. These metrics include: AMOTA, average multiobject tracking accuracy; AMOTP, average multiobject tracking precision; sAMOTA, scaled average multiobject tracking accuracy, which ensures a more linear representation across the entire [0, 1] range of significantly challenging tracking tasks; MOTA, multi object tracking accuracy; MT, mostly tracked trajectories; ML, mostly lost trajectories.

**Baselines tracker**. We've chosen the AB3Dmot tracker from [40] as our baseline. This tracker initially retrieves 3D object detections from a LiDAR point cloud. It subsequently integrates the 3D Kalman filter with the birth and death memory technique to guarantee efficient and resilient tracking performance. It attains state-of-the-art performance while maintaining the fastest speed.

**Table 7: Cooperative Tracking benchmark.**

| Method | AMOTA(↑) | AMOTP(↑) | sAMOTA(↑) | MOTA(↑) | MT(↑) | ML(↓) |
|---|---|---|---|---|---|---|
| No Fusion | 12.87 | 36.22 | 46.27 | 37.34 | 24.92 | 51.03 |
| Late Fusion | 24.56 | 44.16 | 60.91 | 51.55 | 38.00 | 25.38 |
| Early Fusion | 21.73 | 40.99 | 57.57 | 51.90 | 34.96 | 26.93 |
| When2Com[12] | 24.30 | 43.68 | 60.69 | 52.62 | 36.75 | 26.21 |
| DiscoNet[11] | 24.58 | 44.71 | 62.79 | 55.70 | 39.81 | 23.89 |
| V2VNet[9] | 26.19 | 47.43 | 65.81 | **56.58** | 41.63 | 23.83 |
| V2X-ViT[10] | 26.85 | 46.50 | 64.50 | 55.92 | 38.32 | **22.44** |
| CoBEVT[7] | 27.09 | **48.47** | **66.82** | 55.20 | 40.93 | 25.33 |
| Where2com[1] | **28.53** | 46.57 | 66.17 | 55.42 | **41.92** | 23.35 |

## 6 Experiments

### 6.1 Implementation Details

We designate No Fusion as our baseline. To ensure a fair comparison, all models utilize PointPillar as the backbone for LiDAR feature extraction and use 32x feature compression (decompress) to save bandwidth. Among them, for CoBEVT, we only use the FuseBEVT module for feature aggregation without the SimBEVT module. As shown in Section 4, U2U Data has 315K lidar frames. We randomly divided it into training set/validation set/test set according to the ratio of 0.7/0.15/0.15 and finally obtained 220.5K/47.25K/47.25K frames respectively. During the training phase, we randomly designate one UAV as the ego UAV and train each model until achieving optimal task performance. During testing, we evaluate all compared models using a fixed ego UAV. For the tracking task, we utilize the previous three frames along with the current frame as inputs.

### 6.2 Cooperative 3D Object Detection

As shown in Table 5 and Table 6, we comprehensively compare synchronous and asynchronous cooperative 3D object detection models on our U2UData dataset. Compared to the No Fusion method, all cooperative perception methods significantly improve detection performance by at least 22.69%/7.40% (AP@IoU=0.5/0.7). Especially in long-range detection, their performance improves by 11.45× and 8.47× (AP@IoU=0.5/0.7). This is because cooperative perception can greatly alleviate the problems of occlusion, sensor performance degradation, and limited perception range of the single UAV. Compared with the Late Fusion method, the Intermediate Fusion method can improve the detection performance up to 15.04%/33.31%(AP@IoU=0.5/0.7). And compared with the Early Fusion method, the Intermediate Fusion method can improve the detection performance up to 11.59%/19.57% (AP@IoU=0.5/0.7), and reduce the communication cost by 84.15%. This is because the Intermediate Fusion method performs multi-scale feature extraction and integration in the feature encoding module before communication and the spatio-temporal cooperation module after communication. It can improve the performance of the perception task and reduce the communication cost in a fine-grained way.

In the Sync setting, among all the Intermediate Fusion methods, Where2com has the best performance with respect to AP@0.5 and AP@0.7, 0.4% higher than the second best model CoBEVT, 11.59%

higher than Early Fusion, and 15.04% higher than Late Fusion. In the Async setting, Where2com has the best performance against AP@0.5, and V2V-ViT has the best performance against AP@0.7. Except for the No Fusion method, the performance degradation of all methods is very obvious when a communication delay is introduced. The results show that asynchrony in cooperative perception tasks is a key issue that needs to be addressed urgently.

### 6.3 Cooperative 3D Object Tracking

As shown in Table 7, we comprehensively compare the performance of the cooperative perception models in 3D object tracking tasks on our U2UData dataset. Compared to the No Fusion method, AB3Dmot combined with cooperative detection significantly improves the tracking performance by at least 8.86% AMOTA and 11.3% sAMOTA. Compared with the Late Fusion method, the Intermediate Fusion method can improve the detection performance by up to 3.97% AMOTA. Compared with the Early Fusion method, the Intermediate Fusion method can improve the detection performance up to 6.8% AMOTA. Similar to the cooperative detection path, CoBEVT and Where2Com achieve the best performance in most of the evaluation metrics.

## 7 Conclusion

In real-world scenarios, UAVs are extremely sensitive to the effects of smoke, temperature, humidity, and airflow due to their small size. Taking fire rescue missions as an example, the presence of red flames and smoke significantly impairs the performance of visual sensors. Brightness and smoke sensors can enhance visual sensors in detecting targets through filtering. Airflow influences the direction in which the fire spreads, while temperature and airflow sensors assist UAVs in planning paths and guiding survivors to safety. However, because these sensors are closely related to terrain and meteorology and have strong interactions between modalities, existing UAV simulators are difficult to collect, and the model trained by existing simulators is difficult to deploy in the real world. We build U2USim, the first real-world mapping swarm-UAVs simulation environment. We present U2UData, the first large-scale cooperative perception dataset for swarm-UAVs autonomous flight. We hope U2USim and U2UData can assist UAVs algorithms in being deployed in the real world.

## 8 Acknowledgments

This work was supported in part by the National Key Research and Development Program of China No. 2020AAA0106300, National Natural Science Foundation of China (No. 62222209, 62250008, 62102222), Beijing National Research Center for Information Science and Technology (No. BNR2023RC01003, BNR2023TD03006), China Postdoctoral Science Foundation under Grant No. 2024M751688, Postdoctoral Fellowship Program of CPSF under Grant No. GZC20240827, and Beijing Key Lab of Networked Multimedia.

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
