# OpenReview forum: "U2UData: A Large-scale Cooperative Perception Dataset for Swarm UAVs Autonomous Flight"
_acmmm.org/ACMMM/2024/Conference — MM2024 Oral_

### Official Review · Reviewer_tk8J · 2024-05-20

**Rating:** 4
**Confidence:** 4

**Summary:**

he paper introduces the U2UData dataset and the U2USim simulator for autonomous flight of swarm UAVs. It utilizes real geographic data from Yunnan Province in simulations and collects data under various scenarios using eight different sensor types on UAVs. In addition to the simulator and dataset, the paper provides a benchmark of state-of-the-art approaches for collaborative perception on the dataset and validates its effectiveness for 3D detection and tracking tasks.

**Strengths:**

1. Geographic data from the real world provides realistic terrain and elevation variations, making simulations for UAVs highly meaningful.
2. The dataset is unique with wild animals as objects. I think the use case of UAVs collaboratively sensing to search for wildlife very valuable.
3. In simulations, the dataset models several practical and critical issues such as bandwidth limitations, location errors, and asynchronicity.
4. The data offers a variety of scenarios, including changes in weather and lighting conditions. The volume of data is sufficiently large compared to the state-of-the-art.
5. The baselines used in the benchmarks are comprehensive. The results are well presented.

**Limitations:**

1. The dataset includes only three classes: bear, deer, and wolf. Are there other animals included as noise classes? Additionally, do the bear, deer, and wolf classes have variations in shape, size, and color, or are they modeled uniformly? Are the behaviors of these animals modeled, or are they always static? As the core of the perception task, these details should be introduced.
2. Does the modeling of terrain and trees pose obstacles to UAV flight?
3. The article compares existing baselines, but many of these baselines focus on detection tasks, whereas the paper compares those baselines also for tracking. This distinction needs to be clearly explained.
4. CoBEVT is designed for RGB image input, but the experiment setup in the paper appears to focus on LiDAR point cloud data. Specific adjustments should be described.
5. Minor correction: On line 856, "Late Fusion" should be used instead of "Later Fusion."

**Suitability:**

3

---

### Official Review · Reviewer_o212 · 2024-05-24

**Rating:** 5
**Confidence:** 2

**Summary:**

The paper introduces "U2UData," a comprehensive dataset designed for advancing autonomous flight capabilities in swarm UAVs. Collected within the U2USim environment, this dataset comprises an impressive array of data, including 315,000 LiDAR frames, 945,000 RGB and depth frames, and 2.41 million annotated 3D bounding boxes, alongside essential environmental parameters. Moreover, the paper outlines two pivotal cooperative perception tasks: 3D object detection and 3D object tracking. It then proceeds to benchmark various state-of-the-art cooperative perception algorithms against these tasks, providing a robust foundation for further research and development in the field.

**Strengths:**

The paper presents a dataset that clearly must have required a lot of effort and resources to gather. It has the potential to contribute to the community.

The proposed "U2USim" is the first real-world mapping swarm UAVs simulation environment. These contributions address significant gaps in the field and provide valuable resources for further research and development​​.

The use of actual meteorological data from Yunnan Province to map the simulation environment ensures that the dataset is grounded in real-world conditions, enhancing its utility.

The paper presents results for several state-of-the-art methods for the benchmark tasks. This will allow other researchers to test their methods on this new dataset and compare.

**Limitations:**

The dataset uses meteorological data collected over only five days (March 5-10, 2024). This limited temporal scope raises concerns about capturing the full diversity of temperature, humidity, pressure, wind speed, and wind direction variations across different seasons.

Despite providing additional sensor data such as humidity, smoke, and wind direction, the experimental tasks focus solely on 3D object detection and tracking, which are primarily vision-based. The paper should discuss the potential applications and benefits of these additional sensors in practical tasks to fully justify their inclusion in the dataset.

**Suitability:**

3

---

### Official Review · Reviewer_dg6Z · 2024-05-25

**Rating:** 5
**Confidence:** 3

**Summary:**

This paper presents a simulation of a geographical region, Yunnan Province in China, including four terrains and seven weather conditions for simulating the flight of cooperative UAVs. It uses real meteorological data from Yunnan Province provided by the China Meteorological Center to simulate its environment. Consequently, the paper collects a dataset of eight sensory data from three UAVs flying in three types of swarm formation in the simulation. The collected dataset is used to benchmark cooperative 3D object detection and cooperative 3D object tracking across state-of-the-art techniques for each task.

**Strengths:**

* Presentation of a reach dataset collected by three cooperative UAVs in a simulated environment. The dataset includes 8 modalities, including image, LiDAR, humidity, and temperature, among others. To collect the data authors have developed a simulation using Unreal Engine and data provided by China Meteorological Center. This dataset can be beneficial for training UAVs for multi-agent perception tasks.
* The paper is well-written and provides a clear explanation of the different properties of the dataset and simulation. It also includes a good literature review of existing datasets for cooperative tasks.
* The papers use their dataset to compare the state-of-the-art techniques in cooperative 3D object detection and cooperative 3D object tracking tasks.

**Limitations:**

The reviewer does not identify any big-picture limitations of this work. However, addressing the following would strengthen the paper.

* Providing citations for the following parts:
  * Lines 113 to 131 in the related work section.
  * China Meteorological Center.
  * Unreal, Unreal Blueprints.
  * The color stimulus curve of human eye line 404.

* Specifying which sensor modalities are encoded and transmitted to other UAVs? "each UAV first extracts multimodal features through feature encoders and outputs feature maps" Line 624.

* There seems to be a typo in the following lines:
  * Line 688: "The evaluation area extends by [100, 100]m in both..." Is it [0, 100]m?
  * Line 731: "Asynchronous setting: All UAV data transmission delays are fixed at 150s" Is it 150s or 150ms? Also, is the introduced delay fixed for all communications?

**Suitability:**

2

---

### Meta-Review · Area_Chair_iK8z · 2024-06-26

**Recommendation:** Accept (Oral)
**Confidence:** 4

**Metareview:**

This good paper got 2 Accept and 1 WA, so I decide to accept this paper as oral.